

# Genome-wide identification and response stress expression analysis of the *BES1* family in rubber tree (*Hevea brasiliensis* Muell. Arg.)

Bingbing Guo, Hong Yang, Longjun Dai, Xizhu Zhao and Li-feng Wang

Rubber Research Institute, Chinese Academy of Tropical Agricultural Sciences, Haikou, Hainan, China

## ABSTRACT

Brassinolide (BR) plays an important role in plant growth, development, and the adaptation adversity process. Moreover, BRI1-EMS-SUPPRESSOR 1 (*BES1)* genes are crucial transcription factors (TFs) in the BR signaling pathway. To realize the function of HbBES1 family is helpful to improve genetic resources for rubber tree breeding. Based on the rubber tree database, we used bioinformatics to characterize physicochemical properties, gene structure, *cis*-elements, and expression patterns. These results indicated that there were nine *BES1* members in rubber tree, which we named *HbBES1-1* to *HbBES1-9* and divided into two groups (I and II) based on their genetic relationships. *HbBES1* genes in the same group shared similar gene structures and motifs. *Cis*-acting element analysis showed that the promoter sequences of *HbBES1* genes contained many regulator elements that were related to hormone and stress, indicating that *HbBES1* genes might be involved in the regulation of hormone and stress signal pathways. Our analysis of tissue specificity revealed that all of the nine *HbBES1* members expressed highly in branches. Gene expression profiles under different hormone treatments showed that the *HbBES1* gene family was induced to varying degrees under different hormones, *HbBES1-3* and *HbBES1-9* were extremely induced by ethylene (ETH). These results lay the foundation for further exploration of the molecular mechanism of the *BES1* gene family, especially *HbBES1-3* and *HbBES1-9*, regulating plant stress tolerance in rubber tree.

## INTRODUCTION

Brassinosteroids (BRs) are a type of natural plant hormone that are located in plant organs such as pollen, seed, stem, and leaf. They are widely involved in various plant regulation processes, including stem elongation, leaf development, pollen tube growth, xylem differentiation, senescence, photomorphogenesis, and stress response (*Clouse, 1996*; *Thompson et al., 1982*).

Plants are often exposed to a range of biotic and abiotic stresses, resulting in the reduction of their yield and quality. Transcription factors (TFs) play an important role in the regulation of plant growth, development, and stress response by activating or inhibiting the transcription of target genes (*Farrel & Guo, 2017*). TFs also activate a variety of defense

Corresponding author
Li-feng Wang, lfwang@catas.cn

mechanisms in plants that are subjected to both biological and abiotic stresses (*Century, Reuber & Ratcliffe, 2008*). BRI1-EMS-SUPPRESSOR 1 (BES1) is a plant-specific TF that regulates BR signaling. Because BES1 is the homologous protein of BZR1, their amino acid sequences are highly similar (88%), as well as their conversed domain in the N-terminal (97%) (*Wang et al., 2012*), and some studies have referred to it as BES1/BZR1 (*Yu et al., 2018*). BES1 TF is the core participant in BR signal transduction (*Moreno-Risueno et al., 2007*) and plays a decisive role in plant growth, development, and resistance to stress (*Ryu et al., 2010*). BES1 activates the BR signaling pathway through phosphorylation in order to regulate the expression of BR target genes and ultimately regulate the growth, development, and stress tolerance of plants (*Ryu et al., 2010*). In response to BR, BES1/BZR1 accumulated in the nucleus activates biosynthesis of glucosinolates to regulate the gene expression of the BR signaling pathway and promote plant growth and development. Research has shown that BES1 can regulate the BRL3 of BR receptors in the primary tissue of the taproot vascular bundle in *Arabidopsis thaliana*. BR treatment increases the expression level of *BES1/BZR1* and the accumulation in the nucleus, and BES1/BZR1 proteins are negatively regulated by BIN2 kinase (*Salazar-Henao et al., 2016*). BES1/BZR1 TF can be induced by hormones to regulate plant growth and development. Under osmotic stress, the exogenous expression of transgenic plants was less sensitive to abscisic acid (ABA), promoted stem growth and root development, reduced malondialdehyde (MDA) content and relative electrolyte exosmosis rate, and enhanced salt and drought tolerance (*Sun et al., 2020*). A recent study showed that BES1-mediated ABI3 was a negative regulator of flowering and ectopic specified expression of ABI3 led to a severe early flowering phenotype in wild type *A. thaliana* and tomato. This suggested that BES1-mediated ABI3 regulation plays an important role in plant reproductive phase transition with BES1 involved in the auxin signal pathway to mediate plant growth and development (*Hong et al., 2019*). BR increased the size of root meristem by up-regulating PIN7 and down-regulating SHY2, and BES1 could directly bind to the promoter regions of PIN7 and SHY2, which suggested that PIN7 and SHY2 act as direct targets of BES1 to mediate the growth of root meristem induced by BR (*Li et al., 2020*). Additionally, BES1 interacted with the ethylene (ETH) response factor ERF72 to regulate the expression of downstream genes to ultimately affect plant growth and development (*Liu et al., 2018a*; *Liu et al., 2018b*; *Liu et al., 2018c*; *Lv et al., 2018*). BZR1 combined with the E-box of ACO1 promoter to restrain the expression of *ACO4* further influenced ETH production (*Moon et al., 2020*). In rice, OsBZR1 could also bind to the promoter of OsMIR396d to activate its transcription to regulate plant architecture (*Tang et al., 2017*). Currently, there are eight *BES1* family members that have been found in *A. thaliana*, 28 in cole, 22 in cotton, 11 in maize, and six in grape (*Gao et al., 2018*; *Song, Li & Hou, 2013*; *Yu et al., 2018*; *Liu et al., 2018a*; *Liu et al., 2018b*; *Liu et al., 2018c*; *Song et al., 2018*). There have some gene expression pattern analyses of the *BES1* gene family under various biotic and abiotic stresses that have laid a foundation for an in-depth understanding of the family's operational principle mechanism.

Rubber tree is a perennial arbor and typical tropical rainforest plant that is native to the equatorial climate zone of the Amazon River Basin in Brazil. Natural rubber is the only one of the world's top four industrial materials that is renewable. However, in its

existing conditions, it is unlikely that the rubber supply will increase by increasing the planting area of rubber trees. Therefore, improving the yield of rubber trees through genetic improvement is the fundamental measure projected to improve the self-sufficiency rate of natural rubber. It is also an important strategy to improve the economic benefits of rubber planting. In fact, the sequencing of rubber trees provided the foundation for other studies on gene function (*Tang et al., 2016*). Although there is considerable research on TFs in rubber tree, there is no research on the BES1 TF family. This study presents a comprehensive analysis of the BES1 genes in terms of computational identification, phylogenetic relationships, conservation of amino acid residues, gene structures, conserved motifs, *cis* elements, and tissue expression patterns. In addition, we also analyzed the expression of the *HbBES1* gene family in response to abiotic stressors, *e.g.*, ETH, gibberellic acid (GA3), ABA, salicylic acid (SA), jasmonic acid (JA), and brassinolide (BR), at different points in time. These results will help to further clarify the function of BES1 and its response to latex flow in rubber tree.

## MATERIALS & METHODS

### Genome-wide identification of HbBES1 family members and sequential analysis

BES1 amino acid sequences in *A. thaliana* and *Oryza sativa* were downloaded from the Arabidopsis Information Resource (TAIR; https://www.Arabidopsis.org/) and Rice Genome Annotation Project (RGAP; http://rice.uga.edu/). The search for all *HbBES1* genes from rubber tree was executed using HMMER software (*Eddy, 1998*), which was also used to identify the *BES1* gene family in Molecular & Genetic Resources for Hevea tree (HeveaDB, http://hevea.catas.cn/home/index). First, BES1_N HMM model of the domain was downloaded from Pfam (http://pfam.xfam.org/) and BES1_N domain was used to search the rubber tree database using HMMER software and an *E*-value $< 1e^{-5}$ was set. Second, AtBES1 amino acid sequences were used as queries for the genome-wide identification of HbBES1 genes. We removed the redundant and incomplete sequences and then used SMART (http://smart.embl-heidelberg.de/) to confirm the BES1_N domain from the candidate sequences. Sequences with the BES1_N domain were retained for subsequent analyses. We used the ExPASy ProtParam tool (https://web.expasy.org/protparam/) to analyze the HbBES1 family members' physicochemical properties including length, molecular weight (MW), isoelectric point (pI), and grand average of hydropathicity (GRAVY). Cell-PLoc (http://www.csbio.sjtu.edu.cn/bioinf/Cell-PLoc-2/) was used to forecast the subcellular localization of the HbBES1 members.

### Alignment and phylogenetic tree analysis of the HbBES1 family

BES1 amino acid sequences underwent multiple alignment using CLUSTAL X with default parameters. A phylogenetic tree of BES1 from rubber tree, *A. thaliana*, and rice was constructed using the maximum likelihood (ML) method of Molecular Evolutionary Genetics Analysis (MEGA X) (*Kumar et al., 2018*) and assessed by bootstrap analysis with 1,000 replicates for statistical reliability. All of the other parameters were adjusted to the default settings.

## Analysis of gene structure, conserved protein motifs

The gene structure display server 2.0 (GSDS, http://gsds.gao-lab.org/) was used to diagrammatize intron and exon distribution for individual *HbBES1* genes by comparing the cDNAs and genomic sequences from the rubber tree database website (http://hevea.catas.cn/home/index). Multiple EM for Motif Elicitation (MEME, https://meme-suite.org/meme/doc/meme.html) was used to analyze the HbBES1 protein sequences to ascertain the conserved motifs with the following parameters: maximum number of motifs: 15; minimum motif width: 6; maximum motif width: 50; and maximum number of sites: 10.

## Analysis of conserved domain, *cis*-elements, and expression patterns

Clustal Omega3 (https://www.ebi.ac.uk/Tools/msa/clustalo/) and JavaView were used to analyze the HbBES1 amino acid sequences according to multi-sequence alignment and conserved domain. Promoter sequences (2000 bp upstream of ATG) of the *HbBES1* family genes were acquired from HeveaDB in our previous study (*Tang et al., 2016*). *Cis*-elements of each promoter were identified using the PlantCARE server (http://bioinformatics.psb.ugent.be/webtools/plantcare/html/) and we used TBtools to analyze the *cis*-elements of *HbBES1* gene promoters. We obtained RNA-seq (SRP069104) data of CATAS73397 in HeveaDB and TBtools that we used to draw and analyze the expression heat map.

## Plant materials and treatments

To analyze the tissue expression pattern, we harvested root, stem, leaf, flower, branch, and latex samples from 10-year-old healthy trees (CATAS73397) that were planted in the national rubber tree germplasm resource nursery, Danzhou City, Hainan Province, China. Tissue culture seedings from CATAS73397 were planted in Hainan's natural rubber new planting material innovation base. Based on research in 2016 (*Wang et al., 2016*) and our preliminary experiment, we sprayed leaves when the plant height reached 70∼80 cm with a treatment of 200 μmol/L SA, 200 μmol/L JA, 200 μmol/L ABA, 1% (v/v) ETH, 200 μmol/L GA3, and 5 μmol/L BR. Control plant materials were treated with 0.05% (v/v) ethanol. All treated leaves were harvested immediately and frozen in liquid nitrogen. All tissues were stored in −80 °C until used. Each sample included three independent biological replicates.

## Quantitative real-time PCR assay

In this study, six rubber tissues and six treatments were used for qRT-PCR analysis. Total RNA was extracted from different tissues using an RNAprep Pure Plant Plus Kit (TIANGEN, Beijing, China). We used 1 μg of total RNA from different tissues and treatments and the RevertAid RT Kit (Thermo Scientific, Waltham, MA, USA) following the manufacturer's protocols for first-strand cDNA synthesis. For a qRT-PCR assay, the CFX96 Touch system (Bio-Rad, Hercules, California, USA) was used along with ChamQ SYBR Color qPCR Master Mix (Vazyme, Nanjing, China). Each reaction mixture contained 10 μl of ChamQ SYBR Color qPCR Master Mix, 0.4 μl forward primer, 0.4 μl reverse primer, 0.4 μl ROX remixed, 1 μl diluted cDNA, and 7.8 μl nuclease-free water for a total of 20 μl. The qRT-PCR amplification procedure was as follows: stage 1 was initial denaturation for 30 s

at 95 °C, stage 2 was circular reaction for 10 s at 95 °C and 40 cycles of 30 s at 60 °C, stage 3 was melting curve for 15 s at 95 °C, 60 °C for 1 min, and 95 °C for 15 s. *HbActin* (Genbank: hQ260674.1) was used as the internal control. Finally, the average threshold cycle (Ct) was calculated per each sample. The relative expression was calculated using $2^{-\triangle\triangle Ct}$. Three biological repeats and three experimental replicates were carried out for each sample.

## RESULTS

### Identification of *BES1* genes in rubber tree

In this study, we identified a total of nine open reading frames (ORFs) encoding putative BES1 proteins in rubber tree (cultivar: 'CATAS73397') that we named *HbBES1-1* to *HbBES1-9* according to their locus number (Table 1). HMMER was used to identify the *HbBES1* genes, and Pfam and SMART were used to authenticate the conserved domain of BES1. Basic details of the HbBES1 family members, including their biophysical properties such as locus, strand, genomic length, coding sequence, number of amino acids, MW, pI, GRAVY, and subcellular localization, are shown in Table 1. The length of CDS ranged from 865 to 2,113, and the number of amino acids encoded by family members ranged from 287 to 703. The HbBES1-9 sequence was the longest, and HbBES1-4 was the shortest. The members' molecular weight ranged from 30.52 to 79.07 kDa, and the theoretical pI ranged from 5.76 to 9.16. All HbBES1 members were encoded hydrophobic proteins. All HbBES1s were located in the nucleus, HbBES1-4 with HbBES1-9 were also located in the cytoplasm and HbBES1-3 in the chloroplast.

### Characteristic sequence analysis of HbBES1 proteins

To detect the domain sequences and evaluate the conservation of amino acid residues in BES1 domain, multiple sequences in rubber tree were aligned and analyzed using Clustal Omega3 and JavaView software. According to the results (Fig. 1), we found that there was a highly conserved DNA binding domain belonging to an atypical basic helix-loop-helix DNA binding site (bHLH) in the N-terminal. We classified this as the BES1_N domain of BES1 transcription factors in Pfam which can combine with E-box (CANNTG) and BRRE (CGTGT/CG) elements in many BR-induced promoters, except for HbBES1-4. Many S/TXXXS/T sequences of BIN2 phosphorylation sites were identified in the less conserved part of the HbBES1 domain which is crucial for regulating BES1 activities and where the nucleus localization site (NLS) is found. In the HbBES1 family, the PEST domain was forecast to recognize and be degraded by 26S proteasome in order to control protein stability.

### Phylogenetic relationships of the BES1 family in rubber tree

To explore the phylogenetic relationships of the BES1 family, we aligned all nine identified BES1 proteins from the rubber tree genome using ClustalX with 14 AtBES1s and OsBES1s and reconstructed an ML phylogenetic tree using MEGA X software. Based on the topology of the phylogenetic tree, bootstrap values, and our previous study, 23 BES1 proteins from three species were naturally divided into two groups (I and II) (*Liu et al., 2018a*; *Liu et al., 2018b*; *Liu et al., 2018c*; *Wang et al., 2019*) (Fig. 2). Group I was comprised of 18 BES1 TFs

**Table 1 Physical and chemical property of *BES1* family genes in rubber tree.**

| Gene name | Locus | Strand | Full length (bp) | CDS (bp) | Amino acid (aa) | Molecular weight (MW) | pI | GRAVY | Predicted subcellular localization |
|---|---|---|---|---|---|---|---|---|---|
| *HbBES1-1* | scaffold0050 | − | 3978 | 997 | 331 | 35451.45 | 8.38 | −0.618 | Nucleus |
| *HbBES1-2* | scaffold0070 | + | 4479 | 1003 | 333 | 35602.46 | 8.34 | −0.595 | Nucleus |
| *HbBES1-3* | scaffold0115 | − | 2202 | 940 | 312 | 34097.25 | 9.16 | −0.599 | Chloroplast, Nucleus |
| *HbBES1-4* | scaffold0123 | − | 4369 | 865 | 287 | 30524.99 | 6.54 | −0.518 | Cytoplasmic, Nucleus |
| *HbBES1-5* | scaffold0253 | − | 16986 | 2068 | 690 | 76821.04 | 5.76 | −0.434 | Nucleus |
| *HbBES1-6* | scaffold0434 | + | 3265 | 949 | 315 | 34187.21 | 8.72 | −0.608 | Nucleus |
| *HbBES1-7* | scaffold0863 | + | 4069 | 1003 | 333 | 35468.33 | 8.75 | −0.592 | Nucleus |
| *HbBES1-8* | scaffold0916 | + | 2512 | 952 | 316 | 34048.8 | 9.11 | −0.648 | Nucleus |
| *HbBES1-9* | scaffold1194 | − | 9323 | 2113 | 703 | 79071.07 | 5.76 | −0.475 | Cytoplasmic, Nucleus |

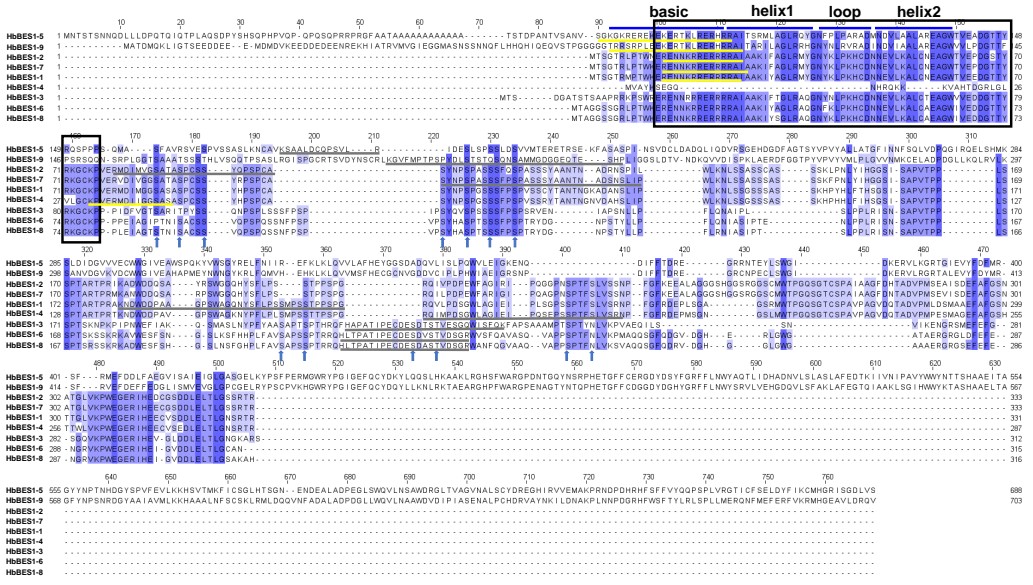

**Figure 1 Multiple sequence alignment of BES1 proteins in rubber tree.** The color of amino acid indicates conservation rate. Conserved amino acids are shaded with different colors. The black box represents N-terminal DUF822, the blue lines signify basic-helix1-loop-helix2 domain (bHLH), the yellow lines represent nucleus location site (NLS), and the gray lines represent PEST domains. Arrows in blue represent S/TXXXS/T sites in nine HbBES1 proteins.

containing seven HbBES1 members, and Group II was comprised of five BES1 transcription factors with two HbBES1s. Each group contained *A. thaliana*, rice, and rubber tree genes, suggesting that the BES1 family in rubber tree had not endured massive expansion because HbBES1 is a conservative family.

## Sequence feature analysis

To comprehensively understand the sequence and structural characteristics of *BES1* genes, gene structure and protein conserved motifs were analyzed. We examined the exon-intron composition in the *HbBES1* family (Fig. 3A). According to their gene structure display, two
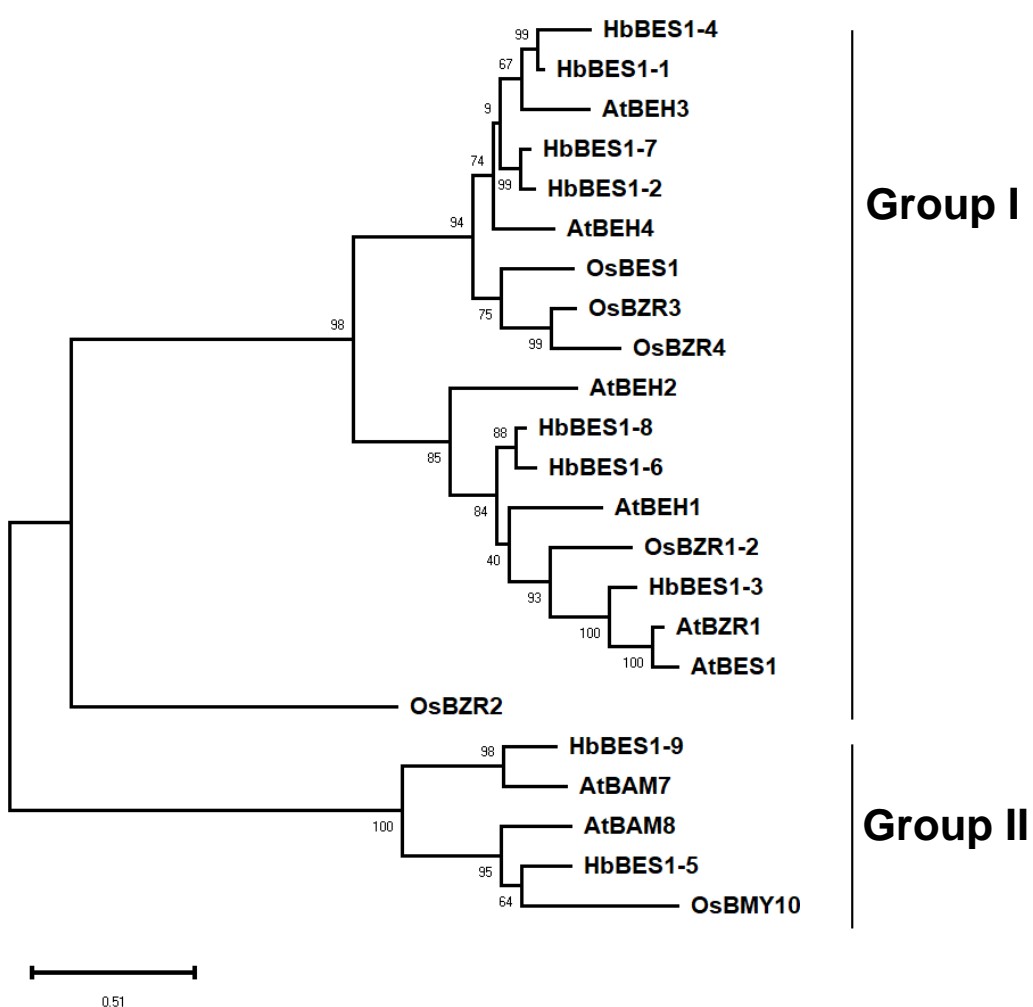

**Figure 2** **Phylogenetic relationships of the BES1 transcription factor family.** Phylogenetic tree was constructed using the ML method with 1,000 bootstrap replications, including eight Arabidopsis BES1 members, six rice BES1 members, and nine rubber tree BES1 members. The bootstrap value of each branch is displayed. The scale bar illustrates 0.51 amino acid substitutions per site.

exons and one intron were located in seven *HbBES1* genes, but *HbBES1-5* and *HbBES1-9* contained 10 exons and nine introns. Furthermore, the manifold of exon and intron lengths showed the diversity of the *HbBES1* family gene structure.

All HbBES1 family protein sequences were analyzed using MEME software to predict and confirm conserved domains, and seven motifs were identified (Fig. 3B) with the accession numbers of the protein sequences shown in Fig. S1. Motif 1 was found in all HbBES1 members except for HbBES1-4, which was the basic helix-loop-helix domain. Motif 2 covered the basic domain in the HbBES1 family except for group II. HbBES1-5 and HbBES1-9 possessed only two motifs and one motif, respectively. The distinct motifs of the HbBES1 family were identified to be consistent with the phylogenetic tree.

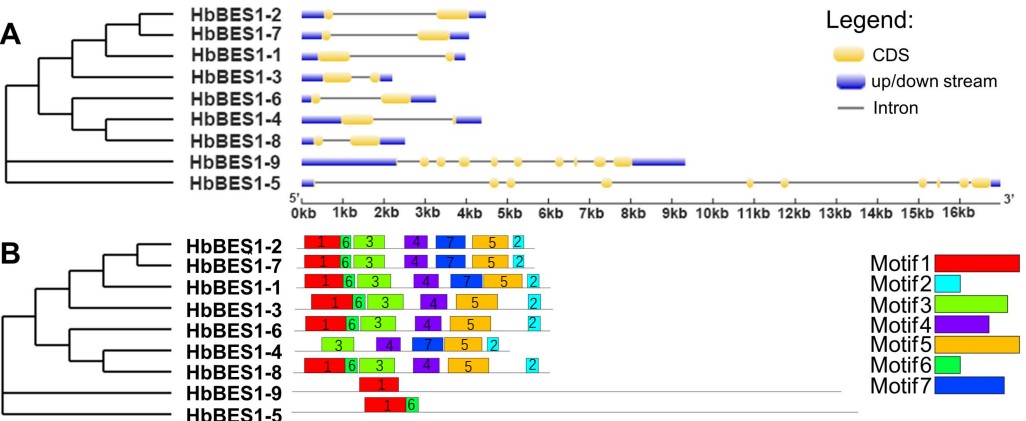

**Figure 3** **Sequence analysis of nine HbBES1 members.** (A) (Left) Phylogenetic analysis of the *HbBES1* gene family constructed using MEGA X with the ML method; (right) gene structures of nine *HbBES1* genes analyzed in GSDS. Introns and exons are represented by lines and yellow boxes, respectively. Blue boxes represent UTRs in the HbBES1 family. (B) (Left) Phylogenetic analysis of the *HbBES1* gene family constructed using MEGA X with the ML method; (right) conversed motifs of HbBES1 family members. Motifs executed using the MEME program are displayed in the panel. Colored boxes with numbers represent different putative motifs.

## Search for *cis*-elements involved in the transcriptional regulation of *BES1* genes

Promoter *cis*-elements played a key role in initiating gene expression and 22 *cis*-elements were identified in the *HbBES1* promoter region that were related to stress response (Fig. 4). *Cis*-acting elements including 3-AF1 binding site, G-box, Sp1, GT1 and MRE (MYB binding site) involved in light responsiveness, GARE, P-box, TATC-box related to GA3 responsiveness, ABRE (an element of ABA responsiveness), AuxRE and TGA-box related to auxin responsiveness, TCA-box related to SA responsiveness, TGACG involved in JA responsiveness, ARE element essential for the anaerobic induction, MYB binding site (MBS) involved in drought inducibility, CAT-box related to meristem expression, MSA involved in cell cycle regulation, and WUN-motif as a wounding responsive element. Additionally, MYB and MYC were analyzed in the HbBES1 prompter sequences and were found to participate in many responses. This showed that *HbBES1* family genes are supposed to participate in different kinds of stress and plant hormone response processes, so the *HbBES1* family is likely involved in plant growth and development *via* transcriptional regulation of hormonal and stress responses. *Cis*-acting element names and functions can be found in Table S2.

## Heatmap of *HbBES1* genes in different tissues and treatments

To investigate the different functions of the *HbBES1* family, the RPKM date from different tissues and treatments of CATAS73397 were downloaded from HeavaDB (Fig. 5) and analyzed by TBtools. According to the results, *HbBES1* genes showed tissue expression specificity, apart from *HbBES1-2*. *HbBES1-7* expressed in rubber tree except in the seed and leaf. Compared to other genes, *HbBES1-3* was highly expressed in different tissues

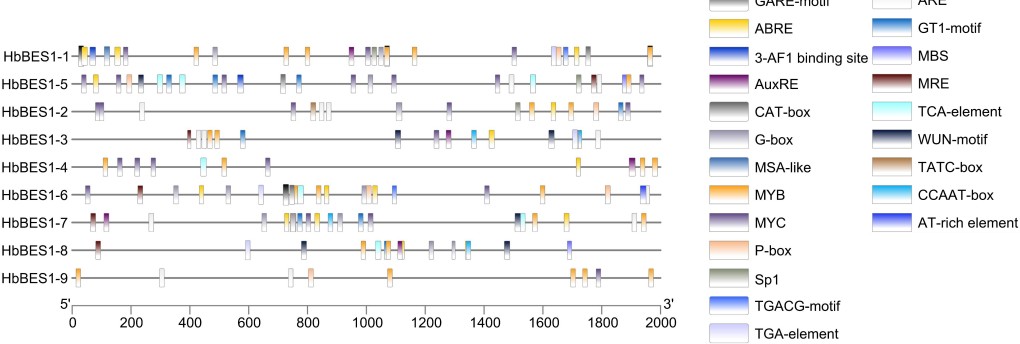

**Figure 4** *Cis*-**regulatory elements of *HbBES1* promoters.** (Left) Distribution of *cis*-elements in HbBES1 promoter sequences. (Right) The boxes with different colors represent different *cis*-elements' responses to various stresses.

of rubber tree, especially in the secondary laticifer, indicating that *HbBES1-3* is perhaps involved in latex formation and flow. *HbBES1-7* was highest expressed in the male flower, indicating that *HbBES1-7* took part in the rubber tree reproduction process. The expression of *HbBES1-8* was balanced in different tissues. Since latex was a crucial yield index of rubber tree, we tested JA and ETH treatments in latex. This showed that the *HbBES1* family had no significant variations under JA and ETH treatments based on RPKM data in HeveaDB. Ultimately, we speculated that BES1 transcription factors of rubber tree probably participated in latex formation, but this should be verified by experimental means.

## Tissue expression specific *HbBES1s* in rubber tree

In order to survey the steady-state patterns of *HbBES1* genes, qRT-PCR was used to determine the specific-expression in root, stem, leaf, flower, branch, and latex with equal amounts of cDNA templates. *HbBES1-2*, *HbBES1-5*, and *HbBES1-8* were widespread expressed in all tissues, and all *HbBES1* genes were highly expressed in flower except for *HbBES1-1* (Fig. 6). *HbBES1-1*, *HbBES1-2*, *HbBES1-7*, and *HbBES1-8* were highly expressed in root. *HbBES1-3* and *HbBES1-4* were only examined in flower. The expression of *HbBES1-4*, *HbBES1-6*, and *HbBES1-9* in latex was almost undetectable. All *HbBES1* genes appeared with various expression patterns in different tissues.

## Expression profiles of the *HbBES1* family response to different hormone stress

BES1 TF was a crucial component in the BR pathway's response to different stressors. Therefore, we performed fluorescence quantitative analysis to detect the expression level of *HbBES1* family members that respond to hormone stress (Figs. 7, 8, 9, 10, 11 and 12). As a whole, the performance of all *HbBES1* genes under ABA, ETH, JA, and SA treatment tended to increase in 6 h, decline in 10 h, and then rise in 24 h, showing fluctuations with the processing schedule. After being treated with GA3 for 0.5 h, all *HbBES1* genes decreased immediately. Under ETH treatment, the relative expression of *HbBES1-3* and *HbBES1-9* manifested drastic change. Strong induction of ABA, ETH, and SA appeared in 6 h, and
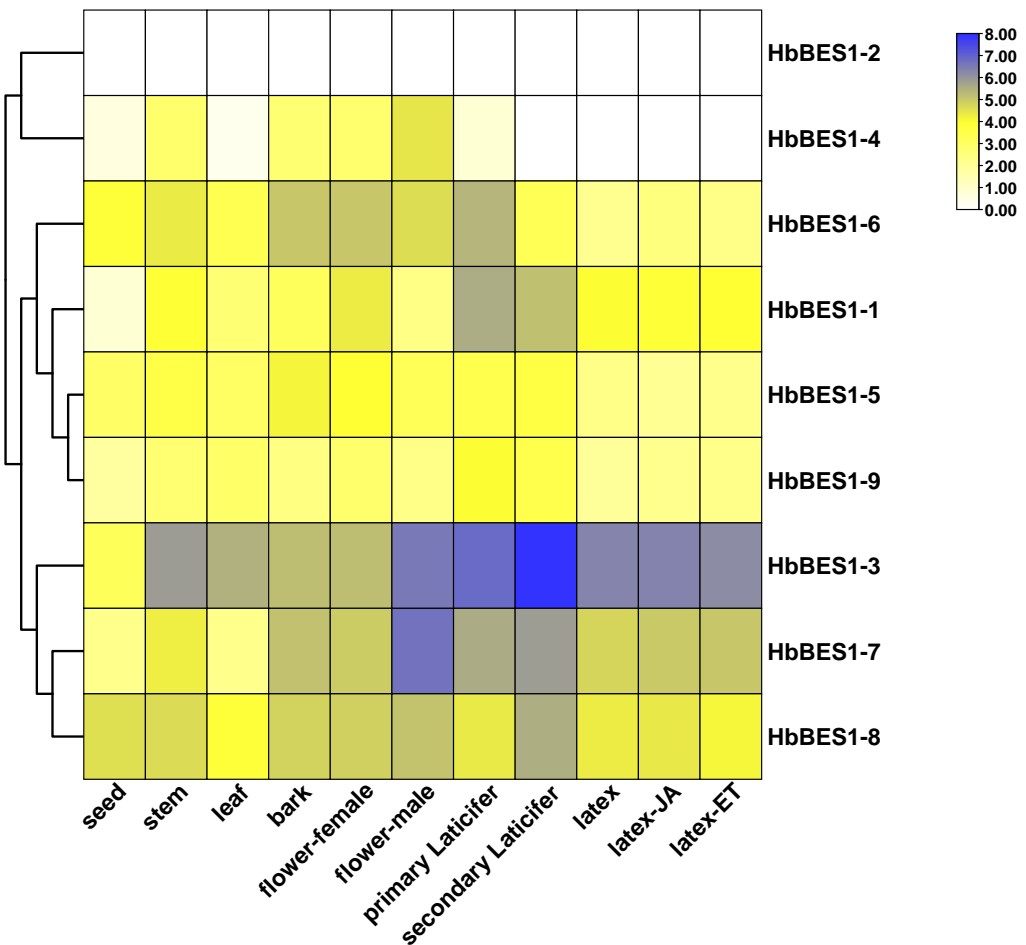

**Figure 5 Expression profiles of *HbBES1* genes in different tissues and treatments.** Based on RPKM data, a heatmap was created in TBtools. Bottom row: different tissues and treatments. Different colors represent different RPKM values. Blue and white represent high and low expression levels, respectively.

JA with BR appeared in 10 h and 2 h. This result indicated that *HbBES1* genes respond to hormone stresses through different patterns.

## DISCUSSION

BR is a highly active hormone that is widely distributed in almost all terrestrial plant tissues. The BR signal transduction pathway starts in membrane receptor BRI1 and co-receptor BAK1, and ultimately ends in BES1/BZR1. The BES1/BZR1 TFs actively participate in plant growth, development, and stress tolerance and are regulated by BR to control tens of thousands of transcriptional networks directly (*Nolan et al., 2017*). Transcriptional regulation is the underlying basis for many of the biological effects of hormones on plants and animals (*Yin et al., 2005*; *Yu et al., 2011*). Some BES1 TFs in *A. thaliana* have been previously studied (*Jiang, Zhang & Wang, 2015*) in order to predict the function of homologous genes in rubber tree.

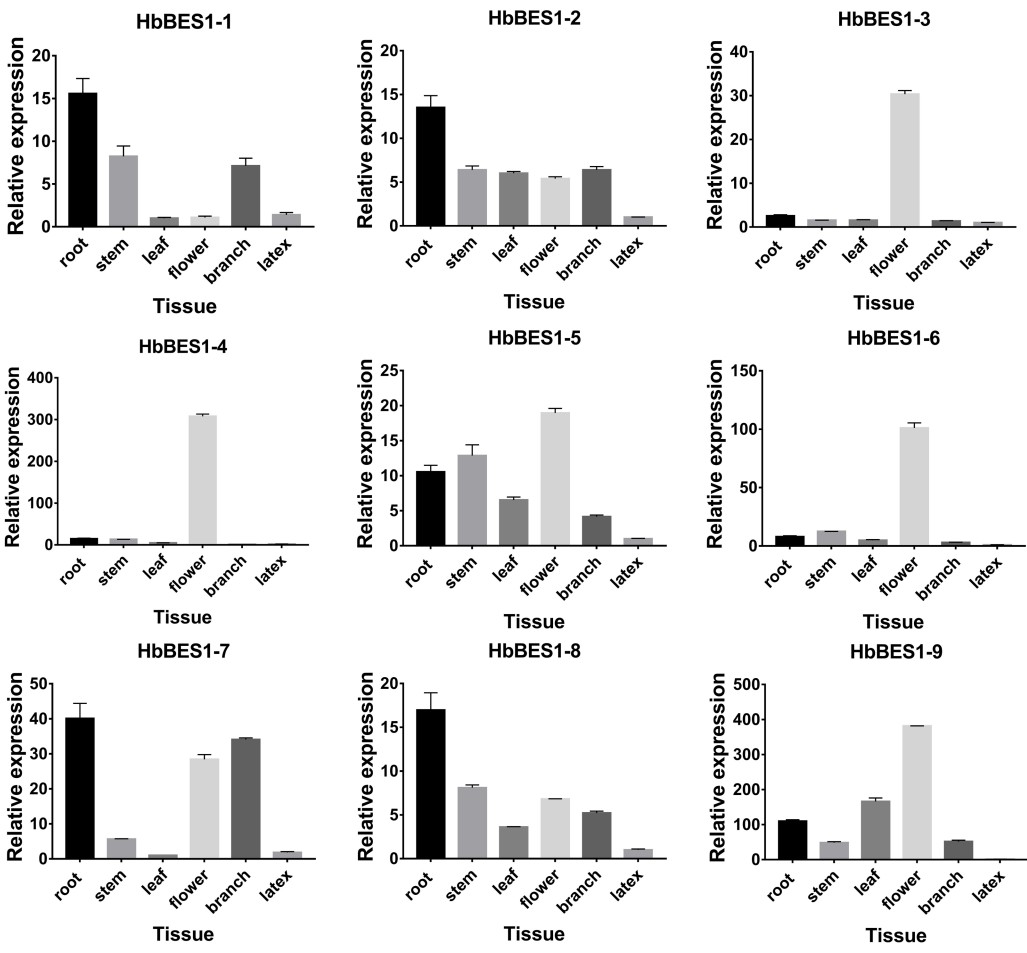

**Figure 6** Specific expression of ni'ne *HbBES1* genes in different tissues (root, stem, leaf, flower, branch and latex). Transcript accumulations were recorded at 35 cycles. *HbActin* acts as reference gene.

The number of BES1 TF in younger plants was fewer than in older plants, suggesting that BES1 plays an important role during plant evolution (*Wu et al., 2016*). Advances in high-throughput sequencing technology have led to the rapid development of multiomics, which provides new databases for the investigation of plant gene families. BES1 TFs have been identified in many species and there are six members in *A. thaliana* that have redundant functions. Rice, maize, Chinese cabbage, and cotton have four, 11, 15, and 11 BES1 TFs, respectively (*Wu et al., 2016*; *Liu et al., 2018a*; *Liu et al., 2018b*; *Liu et al., 2018c*; *Wang et al., 2002*; *Yin et al., 2005*; *Yin et al., 2002*). BES1 in rubber tree, however, has not been studied and exploring their physiological and biochemical functions in rubber tree could provide theoretical basis for breeding new varieties with high yield and high quality. Using bioinformatics methods, we appraised nine HbBES1 putative members in the rubber tree database and comprehensively studied their physicochemical properties, conversed domain, phylogenetic analysis, gene structure, conversed motif, *cis*-element, tissue specific expression, and response to different stimulates.

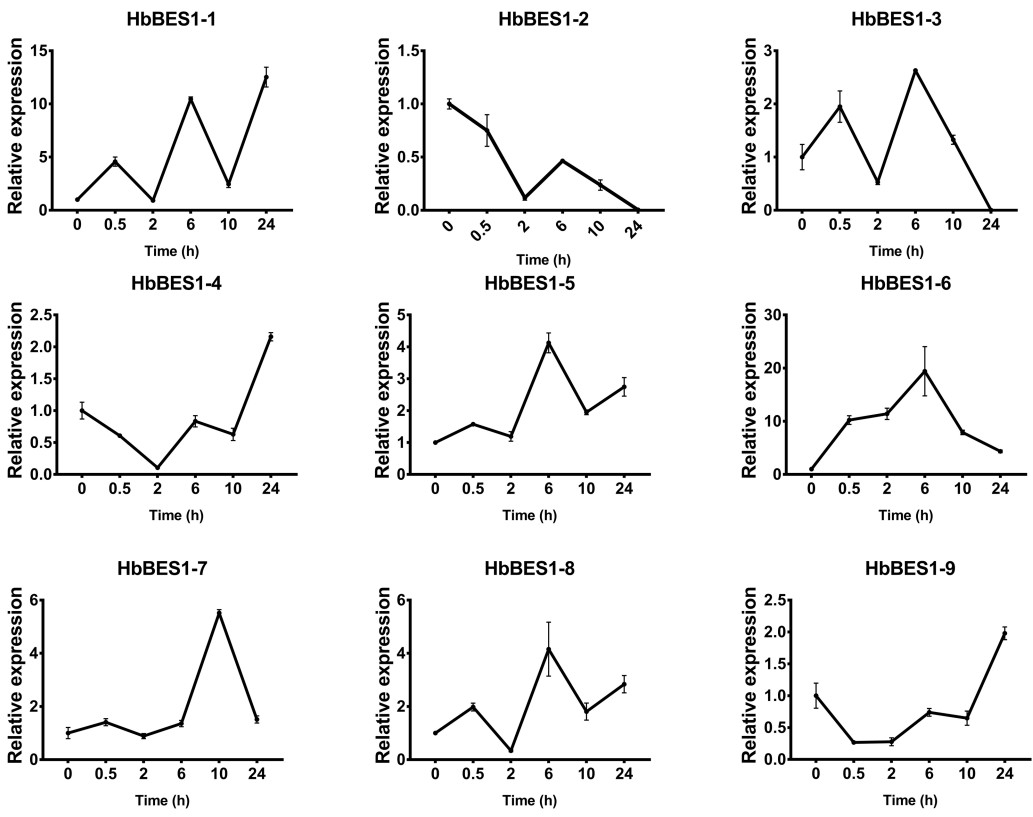

**Figure 7** **QRT-PCR expression analysis induced by ABA.** Expression patterns of HbBES1 genes in response to ABA for 0.5 h, 2 h, 6 h, 10 h and 24 h. *HbActin* was used as an internal reference gene. Error bars represent ± SD of three independent biological repetitions. ABA, abscisic acid.

BR is a stress moderator that is widely distributed in plants to help resist hot, cold, drought, salt, herbicides, insects, pathogens, and other stresses (*Vladimir, Vladimir & Aede, 2000*; *Bajguz & Hayat, 2009*; *Campos et al., 2009*; *Li et al., 2018*; *Sun et al., 2010*; *Yang et al., 2011*; *Yu et al., 2008*). According to our results, we found that the number of BES1 TFs in different species have no significant variations even when there are relationships in different species. Furthermore, the majority of HbBES1 members had pI that were greater than 7, meaning that many of the *BES1* genes in rubber tree were suitable for a saline-alkali environment. Most of the *HbBES1* members with a low instability index were hydrophobic and contained basic amino acids. All HbBES1 members were located in the nucleus, but *HbBES1-4* and *HbBES1-9* were also located in the cytoplasm and *HbBES1-3* was also located in the chloroplast, so we speculated that BES1 TFs play a role in signal transduction in the cell nucleus.

In general, TFs in the same taxonomic group have recent common evolutionary origins and specific conserved motifs that are related to molecular functions, such as forecasting unknown protein functions (*Wu et al., 2016*). A new phylogenetic tree consisting of *A. thaliana*, rice, and rubber tree was reconstructed to explore the evolutionary process and predict the functions of the HbBES1 family. We divided the members into two groups

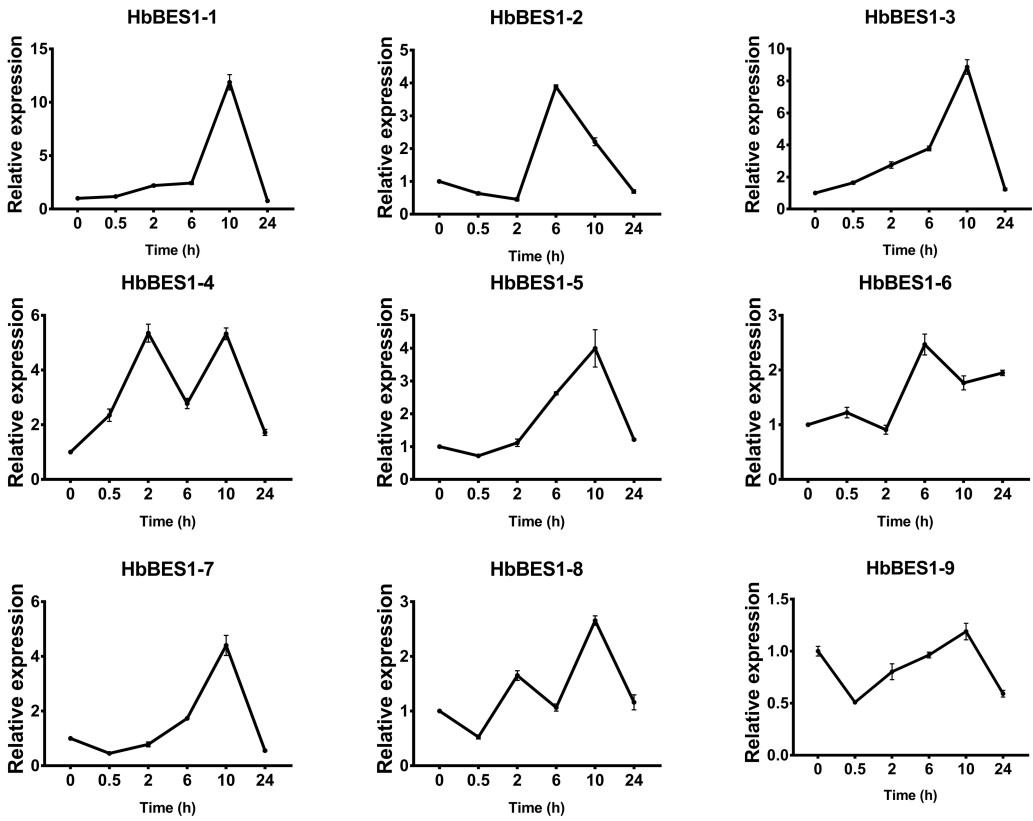

**Figure 8** **QRT-PCR expression analysis induced by BR.** Expression patterns of HbBES1 genes in response to BR for 0.5 h, 2 h, 6 h, 10 h and 24 h. *HbActin* was used as an internal reference gene. Error bars represent ± SD of three independent biological repetitions. BR, brassinolide.

as previously described (*Yu et al., 2018*). Group I contained seven *HbBES1* genes and group II contained two *HbBES1* genes. *HbBES1-3* was the homologous gene of *AtBES1* (AT1G19350.3), *AtBZR1* (AT1G75080.1), and OsBZR1-2 (LOC_Os07g39220). *AtBZR1* was redundantly involved in plant growth, development, and mediation of BR-induced growth and feedback to suppress BR biosynthesis (*Li et al., 2009*; *Saito, Kondo & Fukuda, 2018*; *Wang et al., 2002*), *AtBES1* participated in glucosinolate biosynthesis *via* BR (*Li, 2010*). *HbBES1-6* and *HbBES1-8* clustered with *AtBEHs* that might be involved in related regulation. *HbBES1-5* and *OsBMY10*, *HbBES1-9* and *AtBAM7* were ortholog gene pairs. Under cold stress treatment, OsBMY10 was down-regulated in *osmyb30-oe* plants but up-regulated in the *osmyb30* mutant (*Lv et al., 2017*). AtBAM8 had a BZR1 domain that functioned *via* crosstalk and used BR signaling as a metabolic sensor by activating gene expression to control plant growth and development (*Soyk et al., 2014*). On account of the fact that the retention of duplicates led to discrepancies in different species, the number of genes in each rubber tree group was different when compared to *A. thaliana* and rice.

Through amino acid sequence alignment of HbBES1s, we found that there were a highly conserved bHLH domain and BES1_N in the N-terminal that were similar to BES1 in other known species with DNA binding activity that we proved *via* vivo/vitro tests

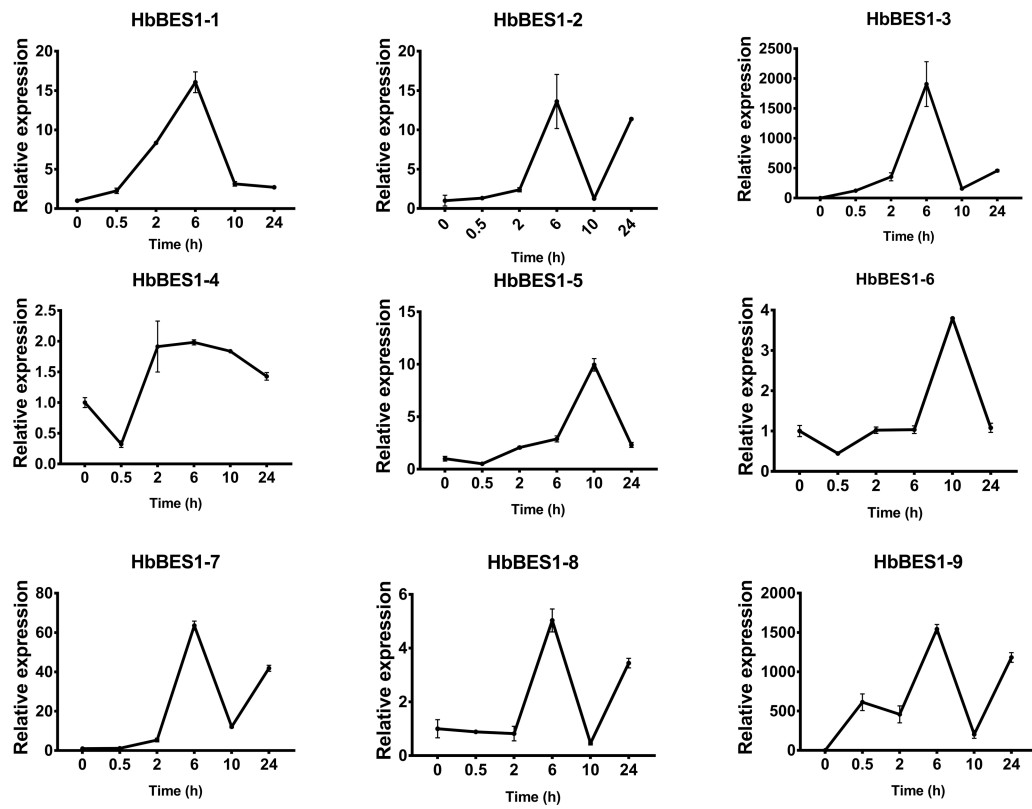

**Figure 9** **QRT-PCR expression analysis induced by ETH.** Expression patterns of HbBES1 genes in response to ETH for 0.5 h, 2 h, 6 h, 10 h and 24 h. *HbActin* was used as an internal reference gene. Error bars represent ± SD of three independent biological repetitions. ETH, ethylene.

(*Yu et al., 2018*; *Zhu et al., 2020*). Phosphorylation and dephosphorylation are important for regulating BES1 transcriptional activity. In the latter part of the HbBES1 protein sequence, there was a S/TXXXS/T domain that was recognized by GSK-3 kinases to be dephosphorylated, which affected interactions in the spatial form of proteins (*Wang et al., 2002*; *Yin et al., 2002*). Gene structure polymorphism has expounded the structural evolution of the gene family (*Roy & Penny, 2007*). In this study, we also identified sequence characteristics of the *HbBES1* family that were similar to other species. The results indicated that the distribution and position of exons and introns of *HbBES1* genes were significantly different. Using a cladogram, we found that group II containing *HbBES1-5* and *HbBES1-9* was the earliest branch in the *HbBES1* family and that these two genes possessed nine introns, which was the maximum number of introns and the other *HbBES1* genes only had 1 intron. The significant difference was attributed to large scale intron losses that occurred during the evolution of the *HbBES1* family. Meanwhile, the allocation of motifs among inter-groups was also observed with intra-groups. Group II possessed at most two motifs in contrast to the intron number, so we concluded that the particular functions of *HbBES1* genes were related to growth and development in rubber tree, although this needs to be verified by further experiments. Our results were consistent with a previous study that

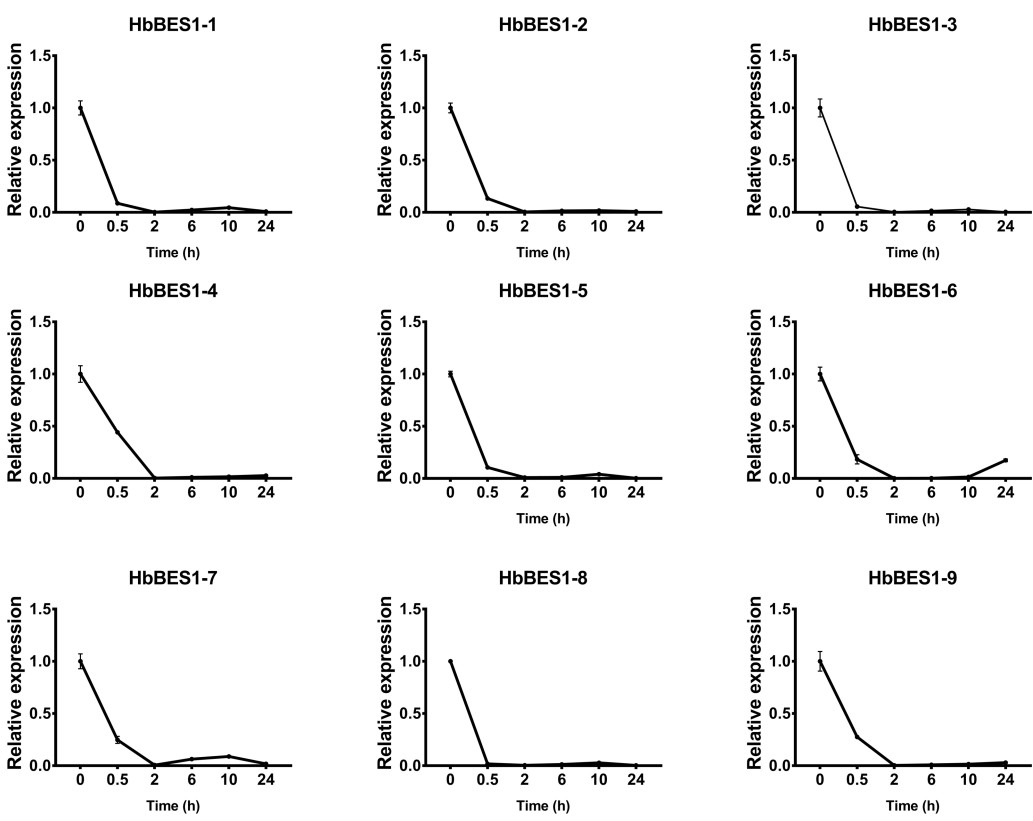

**Figure 10 QRT-PCR expression analysis induced by GA.** Expression patterns of HbBES1 genes in response to GA for 0.5 h, 2 h, 6 h, 10 h and 24 h. *HbActin* was used as an internal reference gene. Error bars represent ± SD of three independent biological repetitions. GA, gibberellin.

found that *HbBES1* genes in the same group shared similar motif distribution patterns, although only one group was comprised of multiple introns while the other two groups had only one intron (*Liu et al., 2018a*; *Liu et al., 2018b*; *Liu et al., 2018c*).

More and more evidence indicates that genes with stress response elements are closely related to environmental changes (*Li & He, 2016*; *Liang et al., 2018*; *Liu et al., 2013*). Based on the results of *cis*-element analysis, we found many stress response elements such as those to ABA, light, and defensive stress. In the *HbBES1* gene family, five *cis*-elements were involved in light responsiveness, illustrating that BR participates in photomorphosis, which has already been verified (*Li & He, 2016*; *Liang et al., 2018*; *Li et al., 2018*). *Cis*-element analysis in *HbBES1* genes showed that the *HbBES1* family is closely affiliated with abiotic and biotic stress involving hormones, for instance, GA3, ABA, SA, auxin, and JA. Simultaneously, *HbBES1* family genes' *cis*-element responses to cell cycle and meristem expression were not only relevant to plant growth and development, but also significantly affected stress resistance.

BES1 TFs are vital regulators in the BR pathway and bind to the promoters that influence different pathways to make a critical difference in plant growth, BR biosynthesis, and abiotic stress (*Belkhadir & Jaillais, 2015*; *Guo et al., 2013*; *Li et al., 2009*; *Wang et al., 2013*; *Wu et*

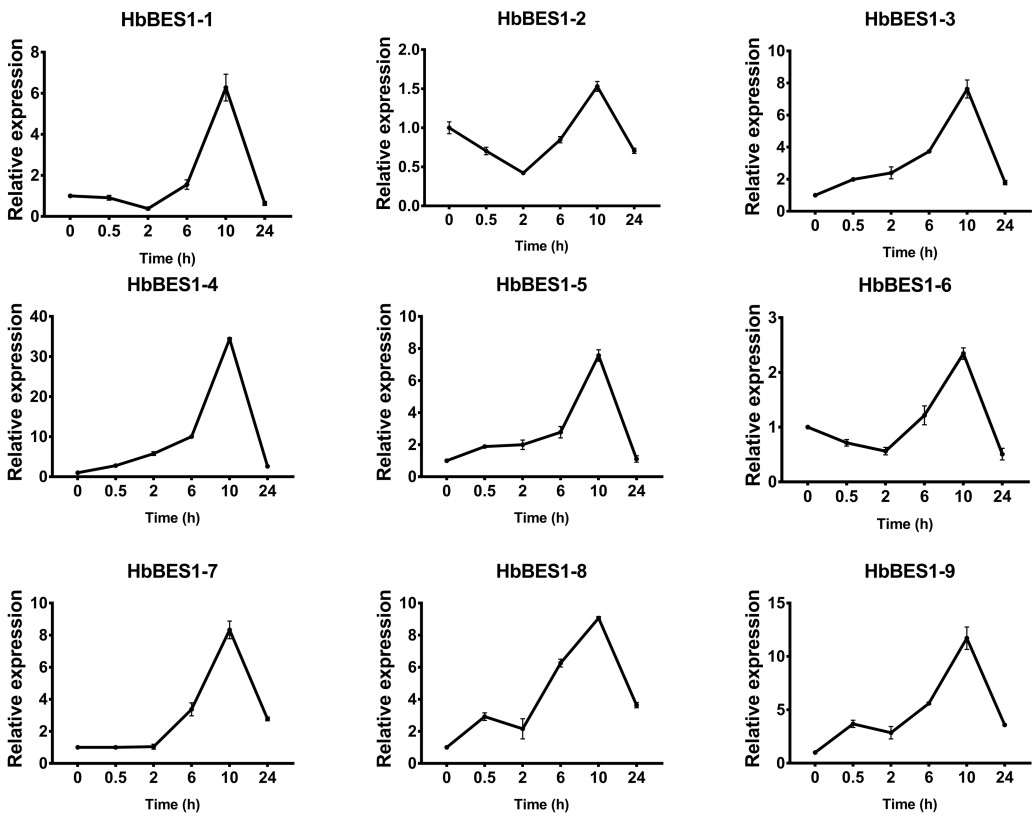

**Figure 11** **QRT-PCR expression analysis induced by JA.** Expression patterns of HbBES1 genes in response to JA for 0.5 h, 2 h, 6 h, 10 h and 24 h. *HbActin* was used as an internal reference gene. Error bars represent ± SD of three independent biological repetitions. JA, jasmonic acid.

*al., 2016*; *Wang et al., 2002*). Combined with microarray analysis and tissue expression in our current genome-wide analysis of rubber tree, we found that their expressions in different organs and different treatments were differential with possible different functions, similar to those found in other species (*Wu et al., 2016*). Several studies have shown that *BES1* genes respond to hormones in distinct ways (*Li et al., 2009*; *Tavakol, 2018*; *Wang et al., 2013*; *Wu et al., 2016*; *Song et al., 2018*; *Wang et al., 2002*). Since *BES1* genes are master regulators of the BR signal pathway (*Guo et al., 2013*), we analyzed the expression patterns of *HbBES1* genes in response to different stresses (ABA, ETH, GA3, JA, SA, and BR). Strong crosstalk had been demonstrated among the stressors. ABA and BR showed antagonistic effects under normal conditions, but endogenous ABA and BR increased in adverse conditions. Additionally exogenous BR increased the expression level of endogenous ABA in order to mitigate the damage caused by adverse circumstances (*Friedrichsen et al., 2002*; *Liu et al., 2011*). Furthermore, a considerable amount of crosstalk between BR and ETH showed that BES1 combines with the E-box element *ACO4* to inhibit *ACO4* expression and to further influence the liberation of ETH (*Moon et al., 2020*). Similarly, BES was associated with the TFs of JA, GA3, SA, and BR (*Bai, Shang & Oh, 2012*; *Bartwal et al., 2013*; *Müssig et al., 2000*; *Cutler, Yokota & Adam, 1991*; *Xia et al., 2014*). Rubber trees are an important

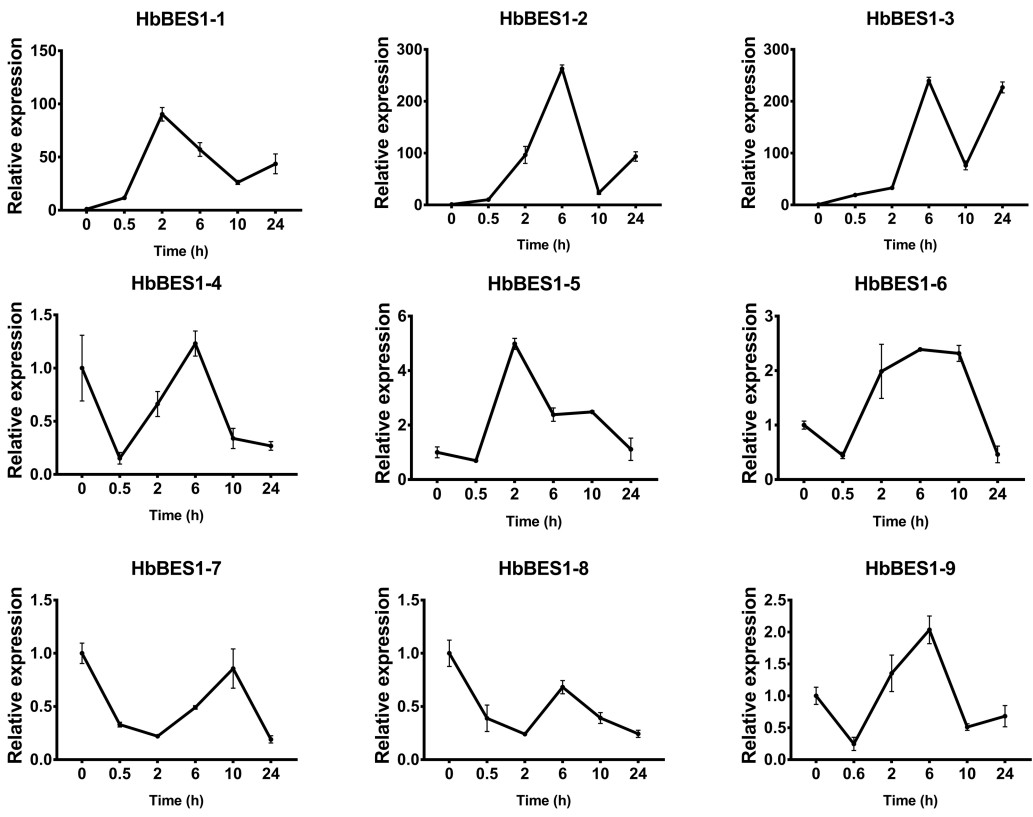

**Figure 12   QRT-PCR expression analysis induced by SA.** Expression patterns of HbBES1 genes in response to SA for 0.5 h, 2 h, 6 h, 10 h and 24 h. *HbActin* was used as an internal reference gene. Error bars represent ± SD of three independent biological repetitions. SA, salicylic acid.

tropical cash crop and latex is a crucial source of natural rubber with high economic value in the fields of industry, defense, and medicine. When rubber trees are mechanically wounded, latex flows out of the laticifers and studies have proved that hormones and stresses can influence latex production (*Gohet et al., 1996*; *Zhang et al., 2021*; *Dusotoit-Coucaud et al., 2010*; *Kim et al., 2003*; *Osborne & Sargent, 1974*; *Zhai et al., 2018*; *Zhao et al., 2011*). In this study, we used different treatments to confirm the response to different hormones and filtrate the regulatory factors of *BES1* genes in rubber tree. The expression levels of the majority of *HbBES1* genes were induced by different hormones. *HbBES1-3* and *HbBES1-9* expression was extremely high when affected by ETH while most *HbBES1* genes under the same treatment showed similar expression tendencies. *HbBES1* family members could be induced by ABA, ETH, GA3, JA, SA, and BR in various ways. The expression levels of *HbBES1* genes all increased under treatments, but decreased in *HbBES1-2* under ABA treatment, all *HbBES1* genes under GA3 treatment, and *HbBES1-7* and *HbBES1-8* under SA treatment. We speculated that the strongest regulatory effects appeared after 6 h with ABA, ETH, and SA; 10 h with BR; 0.5 h with GA3; and 10 h with of JA. Furthermore, we found the putative functions of the highest expression levels of *HbBES1-6*'s response to ABA, *HbBES1-9*'s response to ETH, *HbBES1-4* response to JA, and *HbBES1-3*'s response

to SA were based on *A. thaliana* orthologs's known functions. Therefore, we inferred that HbBES1 TFs have multiple functions and are affected by different hormones during plant growth and development.

## CONCLUSION

In summary, this study provides new perspectives on the function and regulation of the BES1 TF family in rubber tree, and a systematic analysis of the *HbBES1* family incorporating bioinformatics, tissue specific expression, and stress responses. These results will contribute to the selection of appropriate candidate genes for further functional identification, specifically when using molecular breeding techniques to screen potential resources for exploiting stress resistant varieties to abiotic stresses.

## ACKNOWLEDGEMENTS

We acknowledge the HevaDB database platform and its contributors for their meaningful datasets. We also gratefully acknowledge Professor Wang for his help with this paper.

### Funding

This work was supported by the National Key R&D Program of China (2020YFD1001200), Hainan Provincial Natural Science Foundation (Innovative Research Team Project) (321CXTD445). The funders had no role in study design, data collection and analysis, decision to publish, or preparation of the manuscript.

### Grant Disclosures

The following grant information was disclosed by the authors:
National Key R&D Program of China: 2020YFD1001200.
Hainan Provincial Natural Science Foundation (Innovative Research Team Project): 321CXTD445.

### Competing Interests

The authors declare there are no competing interests.

### Author Contributions

- Bingbing Guo conceived and designed the experiments, performed the experiments, analyzed the data, prepared figures and/or tables, and approved the final draft.
- Hong Yang performed the experiments, authored or reviewed drafts of the paper, and approved the final draft.
- Longjun Dai and Xizhu Zhao analyzed the data, authored or reviewed drafts of the paper, and approved the final draft.
- Li-feng Wang conceived and designed the experiments, authored or reviewed drafts of the paper, and approved the final draft.

## Data Availability

The raw data is available in the Supplementary Files.

## Supplemental Information

Supplemental information for this article can be found online at http://dx.doi.org/10.7717/peerj.13189#supplemental-information.

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
