# Peer review of "Genome-wide identification and response stress expression analysis of the BES1 family in rubber tree (Hevea brasiliensis Muell. Arg.)"

_PeerJ, doi:10.7717/peerj.13189_

## Round 0.1 · original submission · Major Revisions

Besides addressing the issues found by our reviewers, please also split Figure 7 in several Figures, in order to increase the resolution and graph size for improved readability.

Reviewer 1 ·

Basic reporting

Based on the rubber tree database, the authors used bioinformatics methods to characterize the physicochemical properties, gene structure, cis-elements and expression patterns of rubber trees. The language of the main text should be substantially improved. There are numerous examples of sentences that are unclear, or even seem contradictory. In some cases, the overall meaning is understandable, but the sentences themselves require rephrasing. As such, a thorough proof-reading is recommended before acceptance. Furthermore, there are some major issues in the manuscript.

Experimental design

In the materials and methods section, the author used three paragraphs to describe RNA extraction, inversion, and final quantification. It is recommended that the author condense this part into a paragraph. Furthermore, for RNA-seq data and HbActin, please note the accession number.
line 123: The author downloaded the HMM model of the domain, not the domain. Please correct.
line 126: The author wrote "based on these two methods." This is very unnecessary. Just write one method for the identification of gene family members, and the other method is just for you to verify whether your results are correct.
In “Plant materials and treatments” section, How to get the concentration used in different disposals, whether to conduct a preliminary experiment or refer to previous studies, please add an explanation.
“Identification of BES1 genes in rubber tree” section lacks description of subcellular location.

Validity of the findings

1. In “Results” section: What is the difference between the research results of “Heatmap of HbBES1 genes in different tissues and treatments” section, and “Tissue expression specificity HbBES1s in rubber tree” section?
The grouping of the gene LOC Os01g08180 in the phylogenetic tree is not accurate, and it is obviously closer to the I and II groups.
Figure 7: it was not cited in the main text. An abscissa title should be included.


2. The discussion section: line 318: the authors concluded that “the number of BES1 transcription factor in different species have no significant variation account for there are relationship in different species.” What does this mean? This sentence does not seem to reveal the relationship between the number of its members and the evolution of the species. The authors should make a collinearity analysis diagram within and between species to reveal the evolutionary relationship more accurately.
Line 291: The author came up with “This result indicated that HbBES1 genes response to stresses through different patterns.”, but used hormones such as ABA, ETH, SA, JA, and even H2O2 in this experiment. The author’s experimental results were too subjective and did not objectively consider the significance of the experimental design.
Lines 336-339: “The number of each group in rubber tree was different with Arabidopsis and rice on account of the retention of duplicates in different groups under different evolutionary constraints was different in Arabidopsis, rice and rubber tree.” This sentence is very confusing. Please reorganize the language expression.
The abbreviated form should be used for brassinolide in discussion.

Additional comments

1. The abstract is very informative but, lacks on developments from a concern. It is best to have this as the opening sentence.
Line 48-50: Please check sentence grammar issues

2. The introduction is well written and has strengths on developing awareness. However, it is best to integrate the components of the proposed aim within the context of literature so that readers can piece the communication into a proposed idea. This helps maintain the reading pleasure as well.
Lines 78, 86: Arabidopsis thaliana? Please check for similar errors in the full text. Furthermore, use the full name for the first occurrence of the Latin name, and replace it with A. thaliana for subsequent occurrences.
There are many grammatical errors in the introduction, such as lines 60, 79, 67-69, 96-99, 89-92 and 109-111. Please re-check this section and correct it. “neGA3tively”, “elonGA3tions”?

Reviewer 2 ·

Basic reporting

In this paper, the BES1 family genes in rubber tree were identified and characterized, including physicochemical properties, gene structure, cis-elements and expression patterns. These results lay the foundation for further studying the function of HbBES1 gene in rubber tree. The results presented here provide interesting data about HbBES1. This manuscript is suitable for publication in PeerJ. However, the manuscript requires some revisions based on the following points.
Some of the statements are not clear and complete. For example, ‘To realize the function of BES1 family in rubber tree, based on rubber tree database, we used bioinformatics to characterize physicochemical properties, gene structure, cis-elements and expression patterns’, etc. The manuscript needs careful editing by someone with expertise in technical English editing paying particular attention to English grammar, spelling, and sentence structure so that the goals and results of the study are clear to the reader.
There are many ‘GA3’ in word, for example ‘Brassinosteroids (BRs) is a natural plant hormone and is located in plant orGA3ns of pollen, seed, stem, leaf and so on to widely involved in various plant regulation processes, such as stem elonGA3tions’, Please check whether it is a spelling mistake or a conversion problem.
References: Some References are incomplete. For example ‘Li T, Kang X, Lei W, et al. SHY2 as a node in the regulation of root meristem development by auxin, brassinosteroids, and cytokinin [J]. Journal of Integrative Plant Biology. 2020, 62(10).’ The reference style should be formatted according to the journal style. Please check carefully, and arrange all of your references according to the standard reference style of the journal.

Experimental design

Plant materials and treatments: More information or details should be given. Tissue culture seedings were treated with SA, etc. How to treat?Spray leaves? ‘All samples harvested immediately frozen’, which tissue? Leaves?

Validity of the findings

‘The molecular weight of members ranged from 30,524.99 to 79,071.07 kDa’, Please check this result.

Additional comments

Quantitative real-time PCR assay: ‘95℃ 15 seconds’, ℃ and seconds should be ℃ and s.
‘CATA S73397’ should be ‘CATAS73397’.
‘ETH (Ethephon)’ and’ Ethylene (ETH)’, I think all should be ETH (Ethephon).

Reviewer 3 ·

Basic reporting

See additional comments

Experimental design

See additional comments

Validity of the findings

See additional comments

Additional comments

1. BES/BZR transcription factors are key components of BR signaling in plants, so that the biological functions of HbBES in BR response should be investigated.
2. AtBES1 and AtBZR1 are homologous gene with high sequence similarity (88%). Their biological functions have many similarities. However, the manuscript is mainly focus on the studies of AtBES1. We think the writers should provide some related data to explain the question.
3. The gene name of BES/BZR transcription factors in figure 2 should be labeled with the published gene name, such as AtBZR1. In addition, the evolutionary relationship of HbBES and OsBZRs also should be analyzed.
4. The Specific expression of 9 HbBES1 genes in different tissues (Figure 6) should be analyzed by qRT-PCR assays, but not semi RT-PCR assay.
5. There are many grammatical mistakes in the manuscript, please correct them

---

## Round 0.2 · Minor Revisions

Your manuscript is almost ready to be accepted. I must, however, ask you to correct Figure 2, which at present does not contain the bootstrap values (unlike the first version, and in contrast to what is written in the figure caption)

Reviewer 1 ·

Basic reporting

no comment

Experimental design

no comment

Validity of the findings

no comment

Additional comments

Thanks for addressing all the revisions and corrections requested.

Reviewer 2 ·

Basic reporting

The authors have revised the manuscript carefully according to the comments and suggestions.

Experimental design

The design of the experiment is reasonable.

Validity of the findings

The results presented here provide interesting data about HbBES1.

Additional comments

The revised manuscript is suitable for publication in PeerJ.

Reviewer 3 ·

Basic reporting

The content of the article meets the requirements of journal publication

Experimental design

The experimental design are reasonable

Validity of the findings

Research results are reliable

Additional comments

No

---

## Round 0.3 · accepted · Accept

Thank you for providing the updated figure. I am glad to accept your paper for publication.